# Listenable Maps for Zero-Shot Audio Classifiers

**Francesco Paissan**[*1,2,5], **Luca Della Libera**[2,3], **Mirco Ravanelli**[2,3], **Cem Subakan**[2,3,4]

[1]Fondazione Bruno Kessler, [2]Mila, Québec AI Institute, [3]Concordia University,
[4]Laval University, [5] University of Trento

## Abstract

Interpreting the decisions of deep learning models, including audio classifiers, is crucial for ensuring the transparency and trustworthiness of this technology. In this paper, we introduce LMAC-ZS (Listenable Maps for Zero-Shot Audio Classifiers), which, to the best of our knowledge, is the first decoder-based post-hoc explanation method for explaining the decisions of zero-shot audio classifiers. The proposed method utilizes a novel loss function that aims to closely reproduce the original similarity patterns between text-and-audio pairs in the generated explanations. We provide an extensive evaluation using the Contrastive Language-Audio Pretraining (CLAP) model to showcase that our interpreter remains faithful to the decisions in a zero-shot classification context. Moreover, we qualitatively show that our method produces meaningful explanations that correlate well with different text prompts.

## 1 Introduction

The widespread adoption of AI in critical decision-making processes makes interpreting the decisions of deep learning models crucial for ensuring transparency and trustworthiness. Recently, significant research has been devoted to explainable machine learning [1]. These efforts aim to either employ interpretable models or explain the decisions of black-box models using posthoc explanation methods. In the audio domain, however, only a few works exist on interpretable audio classifiers [2, 3, 4] as well as on posthoc explanation methods [5, 6, 7, 8]. The latter contributions are limited to standard closed-set classification and do not explore the challenging topic of interpreting zero-shot classifiers. Zero-shot classifiers, on the other hand, are gaining popularity for their exceptional adaptability, as they define audio classes based on a set of textual prompts [9]. The class labels are not necessarily predefined but can be generated dynamically during inference via natural language. The increased flexibility of zero-shot classifiers comes with a drawback: their predictions are challenging to interpret. This difficulty arises from their multi-modal nature, as learning an interpreter in the joint representation space between text and audio is required. A notable example of a zero-shot classifier is Contrastive Language Audio Pretraining (CLAP) [10], which jointly trains audio and text representations using contrastive learning, that we also work with in this paper.

This paper addresses the problem of posthoc explanations for zero-shot audio classifiers. To the best of our knowledge, this has never been attempted before in the literature. Following the masking idea proposed in [8], we propose LMAC-ZS (Listenable Maps for Audio Classifiers in the Zero-Shot context), which consists of a decoder (the interpreter) that outputs a saliency map capable of highlighting the regions within the input audio that trigger the zero-shot classification. We introduce a novel loss function that incentives faithfully following the similarity between the original audio and the corresponding text prompt. Our method provides listenable explanations for linear and non-linear frequency-scale short-time Fourier transform (STFT) representations of audio waveforms. It can also operate on the raw audio domain directly. We applied our explanation method on top of a pretrained version of the popular CLAP [10] by considering different zero-shot classification datasets, including

---

[*]Correspondance to `fpaissan@fbk.eu`

38th Conference on Neural Information Processing Systems (NeurIPS 2024).

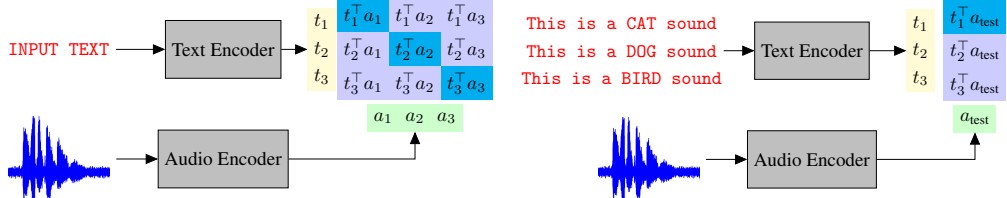

Figure 1: **(left)** The training of the CLAP model for learning cross-modal representations. **(right)** Zero-shot classification with the CLAP model.

the ESC50 [11], UrbanSound8K [12], as well as versions of ESC50 and UrbanSound8K where different types of contaminations are applied. We show extensive experimental results suggesting that the produced saliency maps correlate well with the corresponding text prompts and faithfully follow the original zero-shot classifier. In particular, our evaluation using various faithfulness metrics highlights that LMAC-ZS is able to provide explanations that are highly relevant to the decisions made by the CLAP model in the zero-shot context. Our method significantly outperforms traditional approaches such as GradCAM++ [13], highlighting their inefficiency in challenging tasks such as zero-shot audio classification.

In summary, our contributions are the following:

- We propose a new method, LMAC-ZS, to explain zero-shot audio classifiers.
- We show that LMAC-ZS maintains faithfulness to the CLAP predictions across diverse zero-shot scenarios.
- We qualitatively show that LMAC-ZS produces meaningful explanations for different text prompts.

### 1.1 Related Work

Posthoc explanation methods aim to explain the decisions of pretrained neural networks. Several works exist on producing posthoc explanations with gradient-based approaches in the computer vision literature. These include the standard saliency method [14], GradCAM [15], GradCAM++ [13], SmoothGrad [16], Integrated Gradients (IG) [17], and several others. However, as suggested in [18], these methods often fail to follow the classifier very faithfully and tend to be insensitive even to random model weights. Another category of post-hoc explanation methods in computer vision generates explanations by applying masks to the input data. Key approaches in this category include [19, 20, 21, 22], which use optimization-based techniques to learn and generate these masks. There also exists a series of works that are most closely related to this paper, where a decoder is trained to produce explanations. Notable attempts in this vein include Dabkowski and Gal (2017) [23], Fan et al. (2017) [24], Zolna et al. (2020) [25], and Phang et al. (2020).

In the audio domain, several post-hoc explanation methods exist. These methods employ various techniques such as layer-wise relevance propagation [26], guided backpropagation [27], and LIME [28, 29, 6, 30]. More recent posthoc explanation methods that use a decoder to produce masks on spectrograms include Listen-to-Interpret [5], which uses a Non-Negative Matrix Factorization [31] based decoder to produce non-negative saliency maps. Other examples include Posthoc Interpretation via Quantization [7], which trains a VQ-VAE [32]-based decoder as an explanation module, and Listenable Maps for Audio Classifiers [8], which trains a decoder using a classification loss to promote faithfulness. These works are not directly applicable to zero-shot classification as they require a predefined set of labels to train the interpreter. In this paper, our goal is to produce explanations in a true zero-shot fashion. To achieve that, we train our decoder on the same data as the CLAP model (without using class labels that we will later test on). Subsequently, LMAC-ZS can produce explanations for arbitrary labels, encoded as natural language. This includes labels not previously seen during the training of the interpreter.

## 2 Preliminaries

In this Section, we first present the learning methodology for audio-text cross-modal representations in Section 2.1. Then, we introduce masking-based posthoc explanations in Section 2.2.

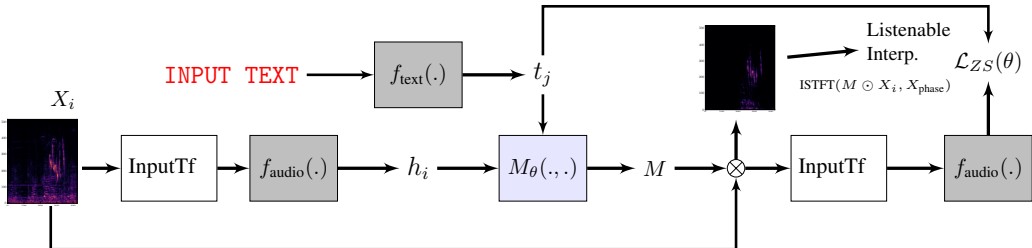

Figure 2: LMAC-ZS architecture. The input spectrogram (linear frequency) $X_i$ (the $i$-th audio in the batch) first of all passes through the transformations (InputTf block) to make it compatible with the input domain (e.g. Mel Spectra) of the audio encoder $f_{\text{audio}}(.)$, which yields the latent representations $h_i$. These representations along with the text representation $t_j$ (the $j$-th text prompt within the batch) are then fed to the decoder $M_\theta(.,.)$. The resulting mask is then element-wise multiplied with the input spectrogram $X_i$. The masked spectrogram $M \odot X_i$ is then converted back to the input domain of the audio encoder, and the similarity score $t_i^\top f_{\text{audio}}\big(M_\theta(t_i, h_j) \odot X_{\text{audio},j}\big)$ is calculated, which is used in the overall training objective $\mathcal{L}_{ZS}(\theta)$. The listenable explanation is produced by simply inverting the masked spectrogram through the inverse-STFT by incorporating the phase spectrogram of the input $X_{\text{phase}}$.

## 2.1 Contrastive Learning of Audio-Text Cross-Modal Representations

The goal of learning audio-text cross-modal representations is to create a joint latent space between text and audio. CLAP (Contrastive Language-Audio Pretraining) [10], achieves this via contrastive learning. That is, the similarity between the latent representations of a text and audio signal is maximized if they form a pair, otherwise this similarity is minimized. More specifically, consider $X_t$ and $X_a$ as batches of text and audio data, respectively. Within the CLAP model, the latent representation is derived by passing the text and audio through their respective encoders, denoted as $g_t(.)$ and $g_a(.)$. This process produces the text and audio latent representations, denoted as $L_{\text{text}} = g_{\text{text}}(X_{\text{text}})$ and $L_{\text{audio}} = g_{\text{audio}}(X_{\text{audio}})$, respectively. Here, $L_{\text{text}}$ is a matrix of dimensions $\mathbb{R}^{N \times T}$, where $N$ is the batch size and $T$ represents the latent dimensionality of text. Similarly, $L_{\text{audio}}$ is a matrix of dimensions $\mathbb{R}^{N \times A}$, where $A$ denotes the latent dimensionality of audio. CLAP trains a joint latent space by passing $L_{\text{text}}$ and $L_{\text{audio}}$ through fully-connected layers such that,

$$t = \mathbf{MLP}_{\text{text}}(L_{\text{text}}), \; a = \mathbf{MLP}_{\text{audio}}(L_{\text{audio}}), \tag{1}$$

where $\mathbf{MLP}(.)$ denotes the multi-layer perceptron transformation layers. The matrix $t \in \mathbb{R}^{N \times d}$ and $a \in \mathbb{R}^{N \times d}$ respectively denote the text and audio latent variables with the same latent dimensionality $d$. As a shorthand for the rest of the paper we will denote the combination of encoders and the MLP with $f_{\text{text}}(.) := \mathbf{MLP}_{\text{text}}(g_{\text{text}}(.))$ and $f_{\text{audio}}(.) := \mathbf{MLP}_{\text{audio}}(g_{\text{audio}}(.))$ for text and audio, respectively. The model aims to maximize the diagonal entries on the matrix $C = ta^\top$. The matrix $C \in \mathbb{R}^{N \times N}$ represents audio-text pairings. The diagonal elements $C_{i,i}$ correspond to positive samples, while other elements are negative samples. This translates into the following training loss function:

$$\mathcal{L}(C) = -\frac{1}{2} \sum_{i=1}^{N} \Big( \log \mathbf{Softmax}_t(C/\tau)_{i,i} + \log \mathbf{Softmax}_a(C/\tau)_{i,i} \Big), \tag{2}$$

where $\mathbf{Softmax}_t(.)$ and $\mathbf{Softmax}_a(.)$ respectively denote Softmax functions along text and audio dimensions, $\tau$ is a temperature scaling parameter, and the $C_{i,i}$ denotes the diagonal elements of the $C$ matrix. We show the training forward pass pipeline in the left panel of Figure 1.

We would like to note that with this framework, the zero-shot classification amounts to calculating the similarity of the representation of a given audio with a set of text prompts, each corresponding to a class labels. Namely, the classification decision is taken as:

$$\widehat{c} = \arg\max_j t_j^\top a_{\text{test}} = \arg\max_j f_{\text{text}}(\texttt{prompt}_j)^\top f_{\text{audio}}(X_{\text{audio}}^{\text{test}}), \tag{3}$$

where $\widehat{c}$ is the zero-shot classification decision, $a_{\text{test}}$ is the embedding for the test audio, and $t_j$ is the text embedding corresponding to the label of class $j$ (represented via $\texttt{prompt}_j$). We show the pipeline of zero-shot classification in the right panel of Figure 1.

## 2.2 Saliency Maps For Fixed Set Audio Classifiers

In this work, we adopt a posthoc explanation method that uses a learnable decoder, following the masking idea introduced in L-MAC [8]. Before we delve into how to generate a saliency map for a zero-shot classifier, we first explain how L-MAC produces a saliency map within the context of a standard classification setup. The loss function that is minimized during training in L-MAC to obtain faithful saliency maps is defined as follows:

$$\mathcal{L}(\theta) = \text{CrossEnt}(\widehat{y}; f(M_\theta(h) \odot X)) - \text{CrossEnt}(\widehat{y}, f((1 - M_\theta(h)) \odot X)) + \lambda \|M_\theta(h)\|_1. \quad (4)$$

The first term in this loss function aims to maximally align the classifier prediction $\widehat{y} = \arg\max_c f_c(X)$, with the classifier output obtained after masking the input, i.e. the logit $f(M_\theta(h) \odot X) \in \mathbb{R}^{N_C}$, where $N_C$ is the number of classes. Note that CrossEnt(.) denotes the CrossEntropy loss function. The decoder network $M_\theta(h)$ takes in the classifier representations $h$ (which can consist of representations of several layers) and produces a mask (with values within the interval $[0, 1]$ and same size as the input) that is element-wise multiplied with the input $X$. A regularization term that consists of an $L_1$ loss is also used to prevent trivial solutions, such as a mask with all values set to 1. The mask-out term $-\text{CrossEnt}(\widehat{y}, f((1 - M_\theta(h)) \odot X))$ minimizes the relevance of the mask-out portion to the predicted class $\widehat{y}$. In the next section, we introduce our framework for explaining zero-shot classifiers (that we have defined in Section 2.1), which again applies a mask to the input to replicate the original text-audio similarities.

# 3 Saliency Maps for Zero-Shot Audio Classifiers

Similarly to the method introduced in L-MAC [8] and summarized in Section 2.2, our goal is to generate explanations that faithfully follow the model. However, in the context of zero-shot classifiers, we do not have a model that outputs a fixed number of logits. Hence, we need a different loss function that promotes faithfulness between the explanations and the zero-shot audio classifier, which relies on similarities to make its decisions. We denote the similarity between the $i$-th text prompt and $j$-th audio recording with $C_{i,j}$ as,

$$C_{i,j} = t_i^\top a_j = t_i^\top f_{\text{audio}}(X_{\text{audio},j}). \quad (5)$$

Our methodology is based on obtaining a saliency map such that the text-audio cross-modal similarity matrix $C$ is maximally preserved after masking the important parts of the spectrogram. In other words, we learn a decoder such that, after masking the audio, the similarity with text prompts within the batch is maximally preserved. To this end, we define the loss function as follows:

$$\mathcal{L}_{\text{ZS}}(\theta) = \sum_{i,j} \left| C_{i,j} - t_i^\top f_{\text{audio}}\Big(M_\theta(t_i, h_j) \odot X_{\text{audio},j}\Big) \right| + \lambda_1 \Big\|M_\theta(t_i, h_j)\Big\|_1 \\ + \lambda_2 \sum_i D(X_{\text{audio},i}). \quad (6)$$

Here, the first term aims to minimize the discrepancy between the original similarities $C_{i,j}$ and the similarities after masking the input audio $X_{\text{audio},j} \in \mathbb{R}^{T \times F}$ using the decoder $M_\theta(t_i, h_j)$, which outputs a mask of shape $T \times F$. Importantly, the decoder is conditioned on both the text representation $t_i = f_{\text{text}}(X_{\text{text},i})$ that corresponds to the $i$-th text prompt in the batch, and the representations $h_j$, which includes the last 4 representations obtained from the audio encoder $f_{\text{audio}}(X_{\text{audio},j})$. $\lambda_1, \lambda_2$ are tradeoff parameters.

The second term in Equation 6 promotes sparsity in the generated mask to avoid trivial solutions. Finally, the last term $D(.)$ aims to increase the diversity of masks generated for a given audio when conditioned on different text prompts. It is defined as:

$$D(X_{\text{audio},i}) = \sum_{j;j \neq i} \left\| t_i^\top t_j - f_{\text{audio}}\Big(X_{\text{audio},i} \odot M_\theta(t_i, h_i)\Big)^\top f_{\text{audio}}\Big(X_{\text{audio},i} \odot M_\theta(t_j, h_i)\Big) \right\|. \quad (7)$$

The goal of this term is to align the uni-modal similarity between text embeddings $t_i$, $t_j$ with the uni-modal similarity between the corresponding audio embeddings $f_{\text{audio}}(X_{\text{audio},i} \odot M_\theta(t_i, h_i))$, $f_{\text{audio}}(X_{\text{audio},i} \odot M_\theta(t_j, h_i))$, obtained from the corresponding masked spectrograms. The intuition

is that the similarity between two text prompts should be reflected in the similarity of the audio embeddings from the corresponding masked spectrograms: the farther the text prompts, the farther apart should be the corresponding audio embeddings from masked spectrograms, and thus, the more different the masks should be. We show the effectiveness of this term on diversity with respect to different text prompts in Section B. The overall pipeline is shown in Figure 2.

**Producing Listenable Explanations:** Our method employs its masking in the linear Short-Time Fourier Transform (STFT) domain, and therefore generating listenable explanations through the inverse-STFT is possible. The listenable explanation is obtained through the following operation,

$$x_{\text{int}} = \text{ISTFT}\left((X \odot M)e^{jX_{\text{phase}}}\right), \tag{8}$$

where both the explanation mask $M$ and the input audio $X$ are in the linear-scale STFT domain, and $X_{\text{phase}}$ is the phase of the original input audio. This operation is also shown in Figure 2.

## 4 Experiments

### 4.1 Metrics

To evaluate our method, we employ faithfulness metrics previously used in the audio interpretability literature for standard classification setups. We adapt such metrics to the zero-shot scenario by using the class prediction probabilities defined by audio-text similarities such that

$$p(c = j) = \frac{\exp(t_j^\top a_{\text{test}})}{\sum_{k=1}^{N_c} \exp(t_k^\top a_{\text{test}})}, \tag{9}$$

where $p(c = j)$ is the probability of predicting the class that corresponds to the $j$-th text prompt and $N_c$ is the total number of text prompts used in the zero-shot setting. Analogously to CLAP [10], to create prompts that correspond to the predefined classes in ESC50 [11] and UrbanSound8K [12], we augment the class labels with the prefix *"this is the sound of"*, obtaining prompts such as *"this is the sound of baby crying"*, *"this is the sound of cat"*. When computing all the metrics for LMAC-ZS, we conditioned the decoder on the text prompt that corresponds to the model prediction $\widehat{c} = \arg\max t_j^\top a_{\text{test}}$.

**Faithfulness on Spectra (FF):** Introduced in [5], it assesses the importance of the provided explanation for the classifier. The metric is calculated by measuring how much does a class-specific prediction probability drops after removing the explanation signal from the original. It is defined as

$$\text{FF}_n = p_{\widehat{c}}(X_n) - p_{\widehat{c}}(X_n - X_{int}),$$

where $\widehat{c}$ is the class prediction given by the classifier. High faithfulness values mean that the masked-in portion of the input spectrogram $X$ is highly influential for the classifier decision of the predicted class $\widehat{c}$. We report the average faithfulness over all examples by reporting the average quantity $\text{FF} = \sum_n \frac{1}{N}\text{FF}_n$. Larger is better.

**Average Increase (AI):** Introduced in [13], it measures the increase in confidence for the masked-in portion of the explanation, and it is calculated as follows:

$$\text{AI} = \frac{1}{N} \sum_{n=1}^{N} [p_{\widehat{c}}(X_n \odot M) > p_{\widehat{c}}(X_n)] \cdot 100,$$

where $[.]$, is the indicator function, which is one if the argument is true, and zero otherwise. For this metric, larger is better.

**Average Drop (AD):** Introduced in [13], it measures the decrease in model confidence when the input image is masked, and it is calculated as follows:

$$\text{AD} = \frac{1}{N} \sum_{n=1}^{N} \frac{\max(0, p_{\widehat{c}}(X_n) - p_{\widehat{c}}(X_n \odot M))}{p_{\widehat{c}}(X_n)} \cdot 100.$$

For this metric, smaller is better.

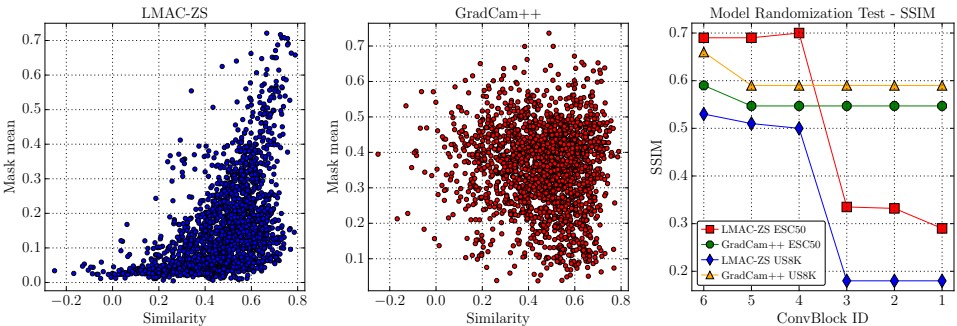

Figure 3: **(left)** Mask-Mean vs Similarity for LMAC-ZS, **(middle)** Mask-Mean vs Similarity for GradCam++, **(right)** Model Randomization Test for LMAC-ZS and GradCam++.

**Average Gain (AG):** Introduced in [33], it measures the increase in confidence after masking the input image. It is calculated as follows (larger is better):

$$AG = \frac{1}{N} \sum_{n=1}^{N} \frac{\max(0, p_{\widehat{c}}(X_n \odot M) - p_{\widehat{c}}(X_n))}{1 - p_{\widehat{c}}(X_n)} \cdot 100.$$

**Input Fidelity (Fid-In):** Introduced in [7], it measures whether the classifier outputs the same class prediction on the masked-in portion of the input image. It is defined as the following and the larger is better,

$$Fid\text{-}In = \frac{1}{N} \sum_{n=1}^{N} [\arg\max_{c} p_c(X_n) = \arg\max_{c'} p_{c'}(X_n \odot M)].$$

**Sparseness (SPS):** Introduced in [34], it measures whether only values with large predicted saliency contribute to the prediction of the neural network. Larger values indicate more sparse/concise saliency maps. We use the implementation from the Quantus library [35].

**Complexity (COMP):** Introduced in [36], it measures the entropy of the distribution of contributions from each feature to the attribution. Smaller values indicate less complex explanations. We again use the implementation from the Quantus library.

### 4.2 Experimental Setup

We use the official pretrained CLAP [10] weights[2] to perform zero-shot classification on ESC50 [11] and UrbanSound8K [12] datasets. We train LMAC-ZS on the datasets on which CLAP had been trained (namely, Clotho [37], FSD50K [38], AudioCaps [39], and MACS [40] which are publicly available). We also explored training LMAC-ZS only on Clotho to simulate the case where the computational budget is limited. The models were trained on a single NVIDIA RTX 3090 GPU. For the LMAC-ZS model that is trained on the Clotho dataset, we did 2 epochs on the complete dataset, for which an epoch approximately takes an hour. For the Full CLAP data we did 2 epochs as well, and an epoch takes around 4 hours. We quantitatively test whether LMAC-ZS follows the zero-shot classifier on In-Domain (ID) and Out-of-Domain (OOD) settings. For the In-Domain setting, we perform zero-shot classification on clean audio from ESC50 and UrbanSound8k and then produce explanations for the classifications using LMAC-ZS. We would like to emphasize that LMAC-ZS has only been trained on the training datasets for CLAP, and has not been fine-tuned on ESC50 or UrbanSound. For the Out-of-Domain setting, we contaminate the audio with various noise sources at 3dB Signal-to-Noise Ratio (other audio from the same dataset, white-noise, and human speech from the LJ-Speech [41] dataset).

We explore masking in the Mel-domain to explore the case where we produce explanations directly in the feature space on which CLAP operates. For Mel-domain we used 44.1kHz data on which the CLAP model is trained. We also explore masking in the linear frequency-scale log power-STFT

---

[2]https://zenodo.org/records/8378278

Table 1: In-Domain quantitative evaluation for the ESC50 and UrbanSound8K Datasets. Two versions of LMAC-ZS are compared: (CT) trained on the Clotho dataset only and (Full) trained on all CLAP datasets. MM denotes the Mask-Mean, the average value for the obtained masks.

| Metric | AI ($\uparrow$) | AD ($\downarrow$) | AG ($\uparrow$) | FF ($\uparrow$) | Fid-In ($\uparrow$) | SPS ($\uparrow$) | COMP ($\downarrow$) | MM |
|---|---|---|---|---|---|---|---|---|
| *ZS classification on ESC50, Mel-Masking, 80.7% accuracy* | | | | | | | | |
| Gradcam | 2.90 | 45.85 | 1.01 | 0.28 | 0.19 | 0.71 | 9.52 | 0.15 |
| GradCam++ | 8.45 | 35.07 | 3.19 | 0.50 | 0.39 | 0.41 | 10.32 | 0.35 |
| SmoothGrad | 0.50 | 52.76 | 0.12 | 0.024 | 0.036 | 0.301 | 10.52 | 0.039 |
| IG | 0.25 | 53.47 | 0.054 | 0.064 | 0.022 | 0.57 | 10.09 | 0.037 |
| **LMAC-ZS (CT)** | **29.00** | **12.25** | **12.93** | 0.49 | **0.80** | 0.78 | 9.40 | 0.14 |
| **LMAC-ZS (Full)** | 23.45 | 17.12 | 10.31 | **0.51** | 0.68 | **0.80** | **9.12** | 0.17 |
| *ZS classification on ESC50, STFT-Masking, 78.9% accuracy* | | | | | | | | |
| GradCam | 20.30 | 23.75 | 7.77 | 0.78 | 0.58 | 0.72 | 11.54 | 0.14 |
| GradCam++ | 32.50 | 8.97 | 7.95 | 0.79 | 0.84 | 0.41 | 12.41 | 0.35 |
| SmoothGrad | 6.95 | 32.75 | 2.85 | 0.78 | 0.47 | 0.53 | 11.98 | 0.0001 |
| IG | 16.10 | 21.51 | 6.05 | **0.79** | 0.65 | **0.74** | 11.58 | 0.0095 |
| **LMAC-ZS (CT)** | 37.40 | 7.43 | **11.26** | 0.78 | 0.86 | 0.50 | **12.29** | 0.11 |
| **LMAC-ZS (Full)** | **43.35** | **4.29** | 10.57 | 0.78 | **0.90** | 0.65 | 11.86 | 0.1 |
| *ZS classification on US8K, Mel-Masking, 71.7% accuracy* | | | | | | | | |
| GradCam | 2.34 | 47.55 | 1.09 | 0.26 | 0.16 | 0.78 | 9.32 | 0.12 |
| GradCam++ | 7.21 | 33.4 | 3.33 | **0.56** | 0.44 | 0.41 | 10.27 | 0.39 |
| SmoothGrad | 1.21 | 49.68 | 0.43 | 0.04 | 0.11 | 0.33 | 10.49 | 0.04 |
| IG | 0.98 | 50.77 | 0.35 | 0.15 | 0.09 | 0.60 | 10.02 | 0.03 |
| **LMAC-ZS (CT)** | 23.41 | 20.58 | 12.88 | 0.51 | 0.65 | **0.85** | 9.01 | 0.08 |
| **LMAC-ZS (Full)** | **35.69** | **15.65** | **18.19** | 0.48 | **0.72** | 0.79 | **8.95** | 0.17 |
| *ZS classification on US8K, STFT-Masking, 68.9% accuracy* | | | | | | | | |
| GradCam | 18.67 | 26.1 | 11.18 | 0.79 | 0.53 | 0.77 | 11.41 | 0.12 |
| GradCam++ | 32.85 | 8.84 | 13.16 | 0.81 | 0.83 | 0.41 | 12.34 | 0.39 |
| SmoothGrad | 15.31 | 23.56 | 7.67 | **0.81** | 0.61 | 0.54 | 11.97 | 0.0001 |
| IG | 22.65 | 19.53 | 12.31 | 0.77 | 0.66 | **0.79** | 11.36 | 0.01 |
| **LMAC-ZS (CT)** | 32.71 | 14.57 | 14.69 | 0.75 | 0.72 | 0.55 | 12.12 | 0.08 |
| **LMAC-ZS (Full)** | **40.85** | **7.79** | **15.52** | 0.78 | **0.85** | 0.76 | **11.34** | 0.07 |

domain to be able to provide listenable explanations. For STFT domain filtering we worked with 16kHz data. We would like to note that this results in slight changes in zero-shot classification accuracies, which are reported in the Tables 1, 2, 3. We trained LMAC-ZS with a batch size of 2 using the Adam optimizer [42] with a learning rate of 1e−5. The decoder consists of a series of transposed convolutions to upsample from CNN14 [43] CLAP representations and incorporates text conditioning by using cross-attention similar to that used in Stable Diffusion [44]. The implementation is done using the SpeechBrain toolkit [45, 46] and it can be accessed through [3].

## 4.3 Quantitative Comparison

We compare LMAC-ZS with popular gradient-based saliency map methods including GradCam [15], GradCam++ [13], SmoothGrad [16], and Integrated Gradients (IG) [17]. We apply these saliency map methods using only the CNN14 audio representations. The class logit with respect to which the class activation map for these methods is calculated is picked by using the zero-shot classification decision $\widehat{c} = \arg\max_j t_j^\top a_{\text{test}}$.

In Table 1, we compare the faithfulness of the explanations obtained on In-Domain data, where we performed zero-shot classification on clean ESC50 and US8k recordings. We observe that on ESC50 with Mel-Domain masking, LMAC-ZS obtains better AI, AD, AG, FF, and Fid-In values. We observe a similar trend for AI, AD, and AG with STFT-domain masking also, while FF values are comparable. On the UrbanSound8K dataset, we also observe that in terms of AI, AD, and AG the best results are obtained with LMAC-ZS trained with the Full CLAP training datasets. In terms of mask sparseness (SPS) and Complexity (COMP) in most cases, the best results are obtained with the proposed model.

In Table 2, we compare the faithfulness of the explanations obtained on ESC50 samples contaminated with three different types of background noises. We observe that with Mel-Masking, LMAC-ZS

---

[3]https://francescopaissan.it/lmaczs

Table 2: Out-of-Domain quantitative evaluation for the ESC50 Dataset.

| Metric | AI (↑) | AD (↓) | AG (↑) | FF (↑) | Fid-In (↑) | SPS (↑) | COMP (↓) | MM |
|---|---|---|---|---|---|---|---|---|
| *ZS classification on ESC50, Mel-Masking, ESC50 contamination, 57.2% accuracy* | | | | | | | | |
| GradCam | 6.78 | 40.71 | 3.13 | 0.29 | 0.19 | 0.69 | 9.66 | 0.18 |
| GradCam++ | 9.82 | 35.81 | 4.53 | **0.42** | 0.29 | 0.39 | 10.40 | 0.35 |
| SmoothGrad | 0.62 | 48.55 | 0.13 | 0.024 | 0.022 | 0.29 | 10.54 | 0.039 |
| IG | 0.55 | 48.88 | 0.091 | 0.073 | 0.020 | 0.56 | 10.13 | 0.039 |
| **LMAC-ZS (CT)** | 19.25 | 24.30 | 8.83 | 0.40 | 0.49 | 0.81 | 9.18 | 0.13 |
| **LMAC-ZS (Full)** | **20.43** | **21.57** | **9.71** | **0.42** | **0.54** | **0.82** | **9.08** | 0.15 |
| *ZS classification on ESC50, STFT-Masking, ESC50 contamination, 58.6% accuracy* | | | | | | | | |
| GradCam | 23.77 | 25.25 | 12.24 | 0.69 | 0.49 | 0.69 | **11.73** | 0.17 |
| GradCam++ | 29.52 | 14.84 | 10.17 | **0.70** | 0.70 | 0.39 | 12.48 | 0.35 |
| SmoothGrad | 11.80 | 30.63 | 5.15 | **0.70** | 0.42 | 0.52 | 12.06 | 0.0002 |
| IG | 16.37 | 25.67 | 7.21 | **0.70** | 0.51 | **0.71** | 11.73 | 0.011 |
| **LMAC-ZS (CT)** | 35.65 | 12.23 | **13.04** | 0.69 | 0.74 | 0.53 | 12.18 | 0.09 |
| **LMAC-ZS (Full)** | **39.4** | **8.28** | 11.81 | 0.69 | **0.80** | 0.67 | 11.79 | 0.09 |
| *ZS classification on ESC50, Mel-Masking, White Noise contamination, 65.2% accuracy* | | | | | | | | |
| GradCam | 3.65 | 43.79 | 1.43 | 0.34 | 0.12 | 0.75 | 9.41 | 0.14 |
| GradCam++ | 7.12 | 37.03 | 2.97 | **0.52** | 0.26 | 0.43 | 10.33 | 0.335 |
| SmoothGrad | 1.72 | 47.93 | 0.56 | 0.040 | 0.040 | 0.28 | 10.54 | 0.035 |
| IG | 1.57 | 47.97 | 0.55 | 0.084 | 0.039 | 0.54 | 10.16 | 0.034 |
| **LMAC-ZS (CT)** | **28.52** | **17.72** | **12.78** | 0.42 | **0.64** | 0.82 | 9.18 | 0.19 |
| **LMAC-ZS (Full)** | 14.25 | 27.92 | 6.62 | 0.41 | 0.42 | **0.86** | **8.86** | 0.11 |
| *ZS classification on ESC50, STFT-Masking, White Noise contamination, 57.4% accuracy* | | | | | | | | |
| GradCam | 14.92 | 31.89 | 5.95 | **0.66** | 0.32 | 0.77 | 11.40 | 0.12 |
| GradCam++ | 19.50 | 24.01 | 8.04 | **0.66** | 0.50 | 0.42 | 12.42 | 0.33 |
| SmoothGrad | 7.10 | 36.53 | 2.66 | **0.66** | 0.25 | 0.52 | 12.15 | 0.0004 |
| IG | 10.17 | 34.35 | 4.89 | **0.66** | 0.30 | **0.69** | **11.80** | 0.011 |
| **LMAC-ZS (CT)** | 19.85 | 21.51 | 7.13 | 0.63 | 0.53 | 0.52 | 12.24 | 0.08 |
| **LMAC-ZS (Full)** | **32.97** | **11.86** | **10.63** | 0.64 | **0.70** | 0.65 | 11.85 | 0.09 |
| *ZS classification on ESC50, Mel-Masking, LJ-Speech contamination, 64.8% accuracy* | | | | | | | | |
| GradCam | 6.50 | 39.05 | 3.06 | 0.33 | 0.20 | 0.70 | 9.66 | 0.18 |
| GradCam++ | 12.85 | 32.81 | 6.50 | **0.47** | 0.32 | 0.41 | 10.36 | 0.35 |
| SmoothGrad | 0.63 | 47.40 | 0.17 | 0.03 | 0.02 | 0.28 | 10.55 | 0.04 |
| IG | 0.53 | 47.70 | 0.10 | 0.10 | 0.01 | 0.56 | 10.12 | 0.04 |
| **LMAC-ZS (CT)** | **24.38** | **20.69** | **11.29** | 0.43 | **0.56** | 0.80 | 9.26 | 0.11 |
| **LMAC-ZS (Full)** | 8.95 | 30.55 | 3.69 | 0.38 | 0.35 | **0.86** | **8.79** | 0.10 |
| *ZS classification on ESC50, STFT-Masking, LJ-Speech contamination, 64% accuracy* | | | | | | | | |
| GradCam | 24.93 | 22.91 | **12.78** | **0.67** | 0.50 | 0.70 | 11.72 | 0.18 |
| GradCam++ | 34.13 | **12.24** | 10.84 | **0.67** | **0.72** | 0.41 | 12.44 | 0.34 |
| SmoothGrad | 9.18 | 29.60 | 3.91 | 0.67 | 0.40 | 0.53 | 12.05 | 0.00 |
| IG | 15.55 | 27.15 | 6.51 | 0.66 | 0.46 | **0.73** | 11.67 | 0.01 |
| **LMAC-ZS (CT)** | **25.77** | 17.79 | 9.67 | 0.63 | 0.63 | 0.61 | 11.96 | 0.04 |
| **LMAC-ZS (Full)** | 25.73 | 15.90 | 7.23 | 0.66 | 0.62 | 0.72 | **11.47** | 0.05 |

reaches better performance in terms of AI, AD, AG, and very comparable numbers in terms of Fid-In. We also observe that in terms of Sparsity and Complexity LMAC-ZS yields better masks in the Mel Domain. In the STFT domain except for LJ-Speech contamination, we observe that LMAC-ZS obtains better performance in terms of AI, AD, and AG. We would like to note that GradCAM++ obtains better FF numbers in general, but we note that GradCAM++ mask areas are larger as shown in the last column with MM. We also observe similar trends for the explanations obtained on US8K samples contaminated with various background noises shown in Table 3. Another point to note is that in general LMAC-ZS trained on the full CLAP training set yields better performance. However, we observe that training LMAC-ZS only on the Clotho dataset yields to comparable or better performance (e.g. ESC50, Mel, white noise contamination). This shows that, in situations where there is limited access to computational resources, training only on Clotho can produce faithful explanations. We furthermore compare the effect of changing the size of the training set size for the interpreter in Appendix C.

## 4.4 Qualitative Comparison and Sanity Checks

We provide some qualitative examples of generated explanations in Figure 4, and compare with GradCAM++ which seems to provide the most faithful explanations among the baselines according

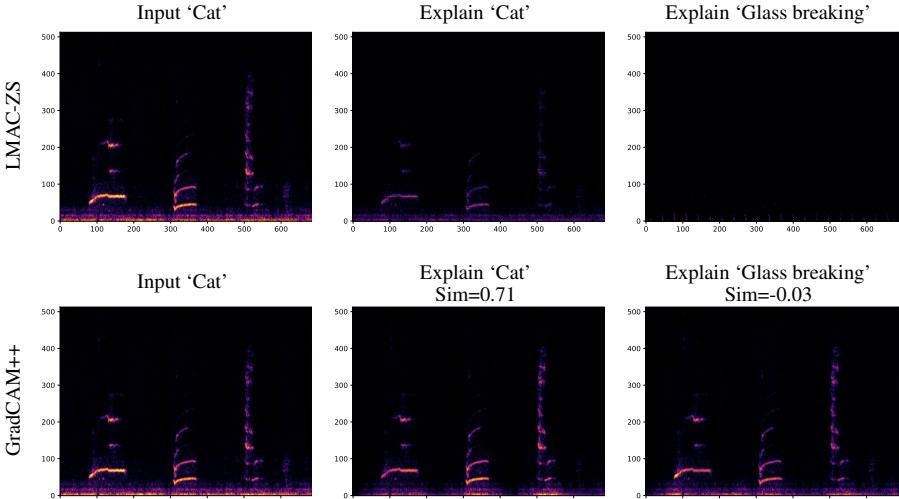

Figure 4: Qualitative Comparisons of Explanations given by LMAC-ZS, and GradCAM++, for two different classes. We see that LMAC-ZS shuts-off the explanation depending on the similarity of the given prompt with the input audio, whereas GradCAM++ remains insensitive to the class label.

to the Tables 1, 2, and 3. We see that LMAC-ZS generates explanations that are much more sensitive to the similarity between the text prompt and the input audio. For instance in LMAC-ZS explanations we see that if there exists a large similarity between the text prompt and the input audio, the mask correctly highlights relevant portions of the input spectrogram. Also, we see that if the similarity between the input and the text prompt is small then the mask tends not to highlight any areas as expected. For instance in Figure 4, we see for the input recordings that corresponds to a 'Cat', both LMAC-ZS and GradCAM++ return reasonable explanations. However, when we prompt LMAC-ZS for an unrelated prompt (e.g. 'Glass Breaking' in this case), it correctly returns an empty explanation mask, as it is impossible to explain. On the contrary, when GradCAM++ returns a class activation map corresponding to the class "Glass Breaking," we observe that the explanation remains unchanged.

To measure the correlation between the mask mean and similarity, Figure 3 presents a scatter plot depicting the relationship between the similarity of the input text prompt and audio. For LMAC-ZS, we observe that explanations are appropriately returned as empty (indicating small Mask-Means) when the similarity score, estimated using CLAP embeddings, is low. Whereas for GradCAM++, the mask mean and similarity appear to be independent of each other.

Finally, we conduct a cascading model randomization sanity check [18] to assess the sensitivity of explanations returned by LMAC-ZS to the CLAP weights. As illustrated in Figure 3, after three layers of randomization, the similarity drastically decreases for LMAC-ZS, while it remains constant for GradCAM++. We visualize these explanations in Figure 5 and provide additional samples in Appendix A.2. More qualitative samples are available through our companion website[4].

## 5    Limitations and Societal Impact

**Limitations**: Our current implementation focuses on fixed-length audio for simplicity. However, the core methodology of LMAC-ZS can be extended to handle variable-length inputs. Additionally, while this work employs standard faithfulness metrics that analyze the dominant class contribution, LMAC-ZS allows for investigating contributions from the top k classes. Studying the top k contributions to faithfulness could provide further insights into the model's decision-making process. Lastly, our study is limited to the CLAP model, primarily selected for its widespread adoption within the field. It is worth mentioning that there is limited availability of alternatives. For instance, most alternative models such as LAION CLAP [47] are still variations of CLAP, offering minimal differences in their core architecture.

---

[4]`https://francescopaissan.it/lmaczs`

**Societal Impact**: We believe this research has the potential for societal benefits, particularly in healthcare applications. While this work does not directly target medical diagnosis, improved explainability of audio classifiers for speech pathologies could make them more trustworthy and accepted by medical professionals. We do not see direct negative societal impacts from this research.

# 6 Conclusions

This paper, to the best of our knowledge, represents the first attempt to develop a model specifically designed for interpreting the decisions of pre-trained zero-shot audio classifiers. In particular, we introduce LMAC-ZS, a novel post-hoc explanation method employing a specialized decoder that generates saliency maps highlighting the regions of the audio input that most contribute to the model predictions. Extensive evaluations highlighted that LMAC-ZS effectively generates explanations that closely align with the decisions made by the CLAP model in zero-shot settings. Our quantitative and qualitative comparisons show that LMAC-ZS outperforms or is comparable to the most popular baseline saliency methods on most quantitative faithfulness metrics. Additionally, LMAC-ZS offers the possibility of being prompted for an explanation. This ability is missing in traditional methods and allows users to gain further insights into the decision-making processes conducted by the model.

## Acknowledgements

This research was enabled in part by support provided by Calcul Québec and the Digital Research Alliance of Canada, and the funds provided by Natural Sciences and Engineering Research Council (NSERC) of Canada.

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

# A  Appendix / supplemental material

## A.1  Results on UrbanSound8K Dataset with Contaminations

Table 3: Out-of-Domain quantitative evaluation for the UrbanSound8K Dataset.

| Metric | AI (↑) | AD (↓) | AG (↑) | FF (↑) | Fid-In (↑) | SPS (↑) | COMP (↓) | MM |
|---|---|---|---|---|---|---|---|---|
| *ZS classification on US8K, Mel-Masking, US8K contamination, 57% accuracy* | | | | | | | | |
| GradCam | 2.64 | 48.43 | 1.43 | 0.27 | 0.12 | 0.77 | 9.42 | 0.13 |
| GradCam++ | 7.58 | 37.89 | 3.91 | **0.56** | 0.33 | 0.37 | 10.39 | 0.40 |
| SmoothGrad | 2.16 | 50.12 | 1.14 | 0.05 | 0.08 | 0.32 | 10.51 | 0.04 |
| IG | 1.82 | 49.79 | 0.82 | 0.18 | 0.07 | 0.59 | 10.06 | 0.03 |
| **LMAC-ZS (CT)** | 17.74 | 25.57 | 9.87 | 0.48 | 0.55 | **0.86** | **8.95** | 0.07 |
| **LMAC-ZS (Full)** | **36.08** | **16.98** | **19.23** | 0.47 | **0.69** | 0.77 | 9.00 | 0.19 |
| *ZS classification on US8K, STFT-Masking, ESC50 contamination, 57% Accuracy* | | | | | | | | |
| GradCam | 17.83 | 31.78 | 12.05 | 0.78 | 0.42 | 0.76 | 11.51 | 0.13 |
| GradCam++ | 28.81 | 14.56 | 14.42 | 0.78 | 0.73 | 0.37 | 12.48 | 0.39 |
| SmoothGrad | 23.13 | 20.58 | 13.73 | **0.79** | 0.64 | 0.52 | 12.12 | 0.0002 |
| IG | 21.53 | 22.41 | 12.76 | 0.74 | 0.60 | 0.77 | 11.53 | 0.01 |
| **LMAC-ZS (CT)** | 31.09 | 17.69 | 15.29 | 0.72 | 0.66 | 0.55 | 12.12 | 0.08 |
| **LMAC-ZS (Full)** | **39.42** | **11.53** | **17.51** | 0.75 | **0.78** | **0.78** | **11.23** | 0.06 |
| *ZS classification on US8K, Mel-Masking, White Noise contamination, 62% accuracy* | | | | | | | | |
| GradCam | 6.77 | 44.01 | 3.91 | 0.35 | 0.21 | 0.73 | 9.46 | 0.16 |
| GradCam++ | 12.51 | 37.77 | 8.49 | **0.60** | 0.31 | 0.38 | 10.38 | 0.39 |
| SmoothGrad | 3.55 | 49.01 | 1.60 | 0.04 | 0.11 | 0.31 | 10.52 | 0.03 |
| IG | 2.51 | 48.43 | 0.94 | 0.08 | 0.13 | 0.56 | 10.11 | 0.03 |
| **LMAC-ZS (CT)** | **42.70** | **12.02** | **25.78** | 0.42 | 0.76 | 0.87 | 8.91 | 0.07 |
| **LMAC-ZS (Full)** | 34.53 | 14.13 | 20.32 | 0.39 | **0.80** | **0.88** | **8.72** | 0.08 |
| *ZS classification on US8K, STFT-Masking, White Noise contamination, 61.1% accuracy* | | | | | | | | |
| GradCam | 18.24 | 35.12 | 12.24 | **0.76** | 0.34 | **0.74** | **11.48** | 0.15 |
| GradCam++ | 20.16 | 27.33 | 13.21 | **0.76** | 0.49 | 0.38 | 12.48 | 0.38 |
| SmoothGrad | 21.36 | 27.98 | 14.25 | **0.76** | 0.47 | 0.52 | 12.21 | 0.0004 |
| IG | 19.91 | 33.36 | 13.74 | 0.72 | 0.36 | 0.69 | 11.79 | 0.01 |
| **LMAC-ZS (CT)** | 27.78 | 17.64 | 13.44 | 0.69 | 0.66 | 0.59 | 12.05 | 0.07 |
| **LMAC-ZS (Full)** | **46.51** | **9.95** | **25.28** | 0.69 | **0.81** | 0.70 | 11.60 | 0.06 |
| *ZS classification on US8K, Mel-Masking, LJ-Speech contamination, 44.9% accuracy* | | | | | | | | |
| GradCam | 3.49 | 46.48 | 1.69 | 0.28 | 0.14 | 0.68 | 9.68 | 0.19 |
| GradCam++ | 10.86 | 36.61 | 6.28 | **0.45** | 0.32 | 0.37 | 10.39 | 0.41 |
| SmoothGrad | 2.04 | 50.09 | 1.10 | 0.03 | 0.05 | 0.31 | 10.35 | 0.04 |
| IG | 1.69 | 49.80 | 0.74 | 0.12 | 0.05 | 0.60 | 10.03 | 0.03 |
| **LMAC-ZS (CT)** | 25.78 | 23.54 | 17.43 | 0.37 | 0.55 | 0.86 | 8.93 | 0.07 |
| **LMAC-ZS (Full)** | **36.24** | **13.90** | **20.47** | 0.41 | **0.73** | 0.86 | **8.79** | 0.10 |
| *ZS classification on US8K, STFT-Masking, LJ-Speech contamination, 46.1% accuracy* | | | | | | | | |
| GradCam | 21.48 | 28.71 | 14.13 | **0.76** | 0.45 | 0.69 | 11.74 | 0.19 |
| GradCam++ | **38.74** | **11.53** | 17.95 | **0.76** | **0.76** | 0.37 | 12.47 | 0.40 |
| SmoothGrad | 34.35 | 19.43 | 24.32 | **0.76** | 0.62 | 0.52 | 12.11 | 0.00 |
| IG | 34.57 | 20.43 | **26.10** | 0.69 | 0.60 | 0.74 | 11.59 | 0.01 |
| **LMAC-ZS (CT)** | 35.96 | 15.91 | 18.33 | 0.68 | 0.67 | 0.63 | 11.92 | 0.07 |
| **LMAC-ZS (Full)** | 32.51 | 13.79 | 15.77 | 0.72 | 0.74 | **0.79** | **10.99** | 0.02 |

## A.2  Qualitative Analysis of Model Randomization Test

Figure 5 presents a qualitative visualization of Model Randomization Test results for GradCAM++ and LMAC-ZS.

## A.3  Qualitative Comparison with GradCAM++

Figures 6, 7 show an additional sample for the quality of the explanations on spectra.

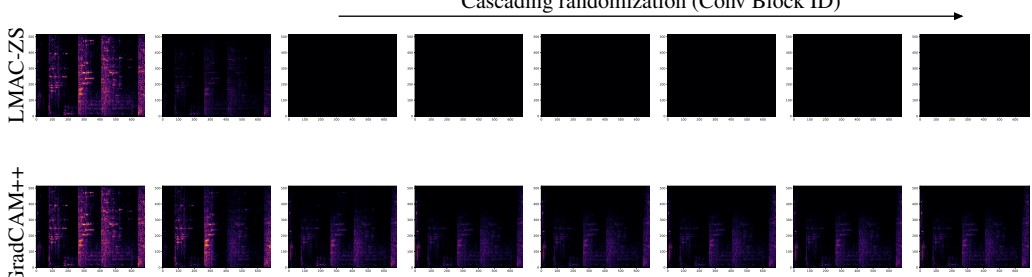

Figure 5: Visualization of Explanations after Cascading Model Randomization. Left column is the input, second column is the original explanation, and more we go towards the right more layers are randomized. Top row is for LMAC-ZS, and the bottom row is for GradCAM++.

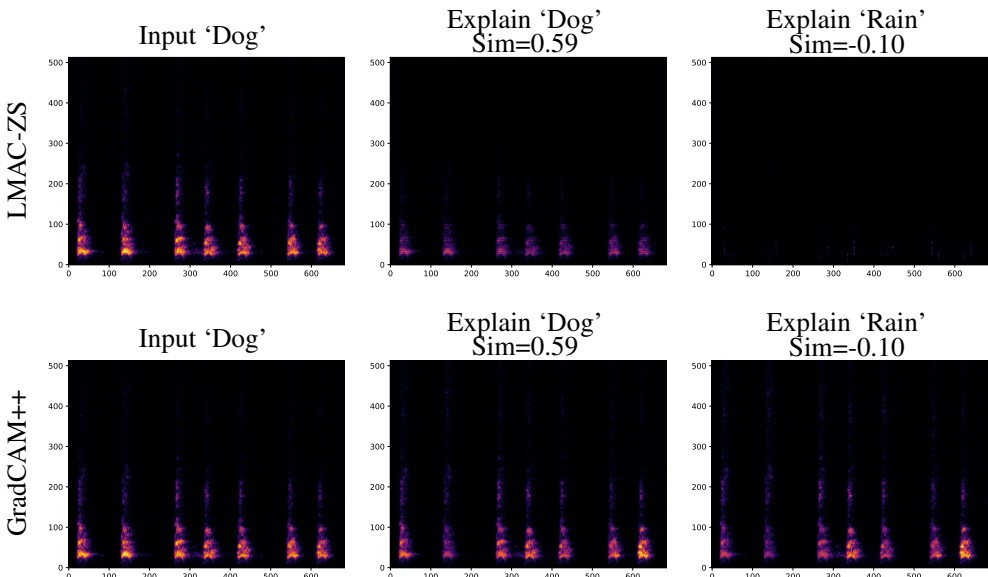

Figure 6: Qualitative Comparisons

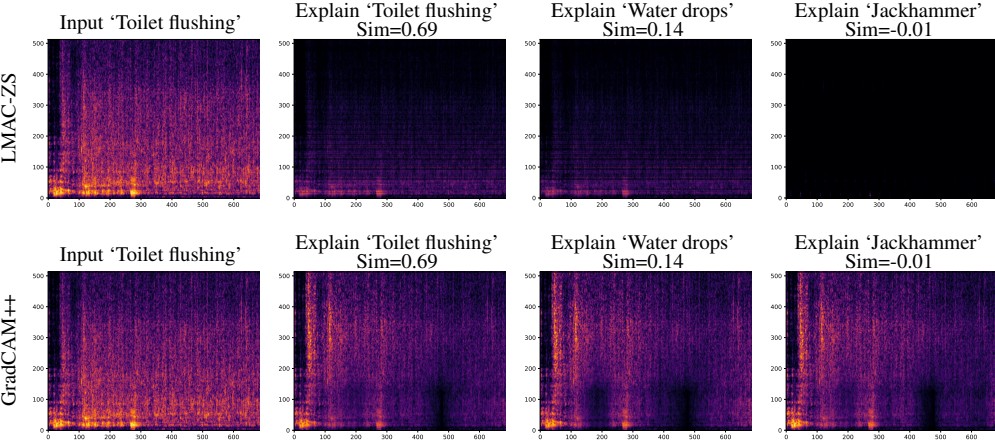

Figure 7: Qualitative Comparisons 2

# B    Explanation sensitivity to Audio-Text Similarity

To showcase the effectiveness of the additional diversity term (Equation 7), we conducted qualitative and quantitative tests to evaluate the explanation sensitivity to text prompts.

## B.1    Qualitative results

We compare the explanations obtained with LMAC-ZS with and without the diversity term in Eq. 7. We present the results in Figure 8 and Figure 9, respectively. The results are obtained for the model that does masking in the STFT domain and was trained on FSD50K. We present the plots using log-frequency scaling. In each plot, we give the original text-audio similarity (in the title of the first subplot), as well as the audio-text similarity after masking the audio (in the title of the third subplot). Note that the predicted class is also used as the prompt for the masking model. We observe that masks are more sensitive to text prompts because of the additional diversity term.

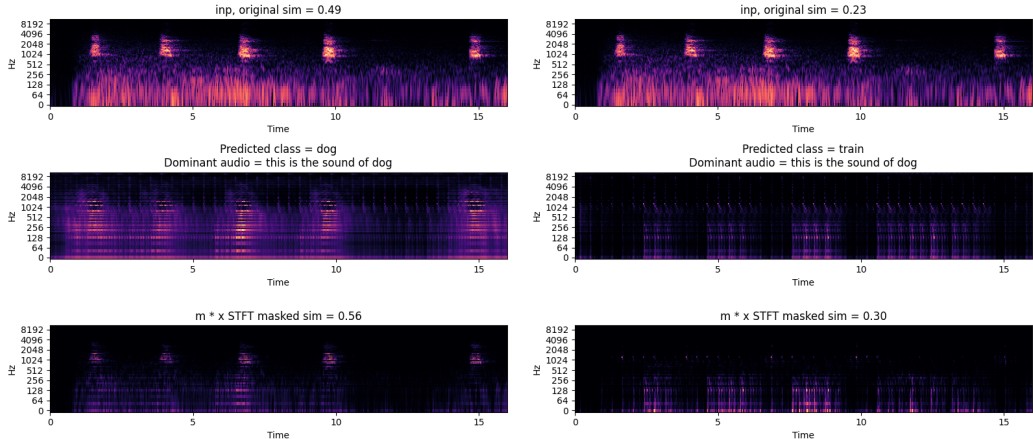

Figure 8: Explanations obtained with the additional diversity term (Eq 7).

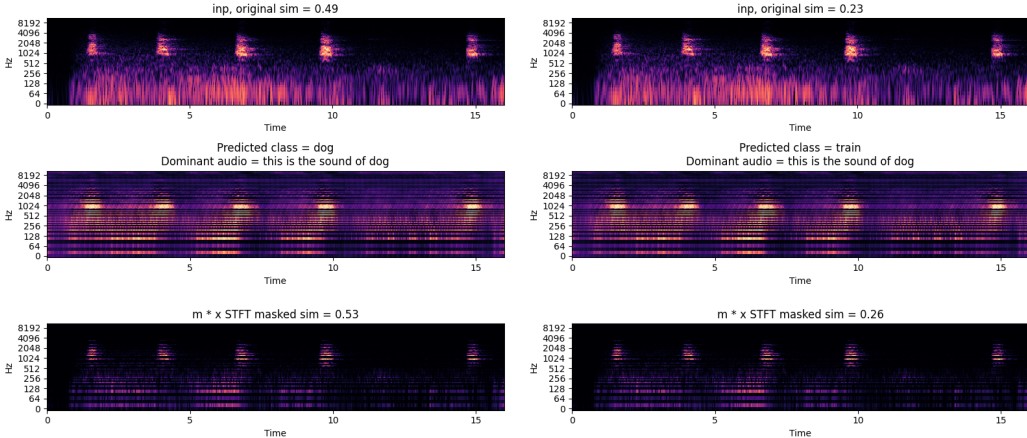

Figure 9: Explanations obtained without the additional diversity term (Eq 7).

## B.2    Quantitative results

In Figure 10, we present the 2D histogram of mask mean and similarity between text and audio after masking. This highlights the increased mask sensitivity of the model to different text prompts when the diversity term in Equation 7 of the paper is utilized. We see in the left panel of Fig. 10 that without the diversity term, the mask means do not have a discernible correlation with the text-masked audio similarity.

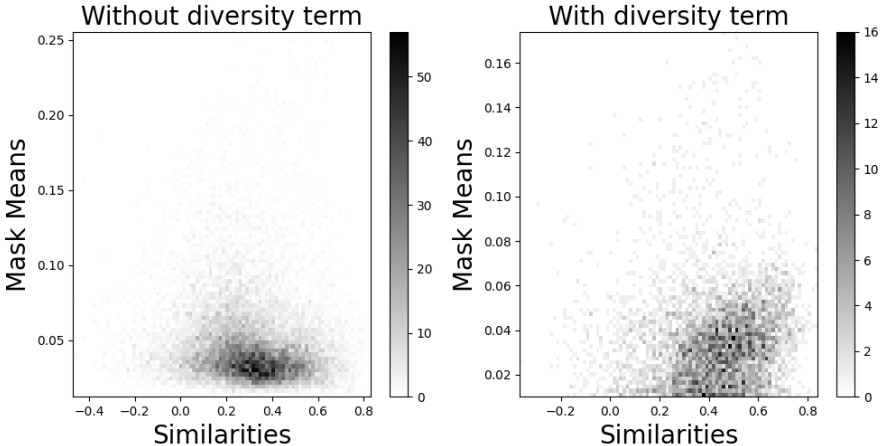

Figure 10: Audio-text similarity after audio masking, without the diversity term (left), with the diversity term (right).

# C   Ablation on the training dataset size for LMAC-ZS

Table 4: Interpreter performance for different training dataset sizes and for additional baselines.

| Metric | AI (↑) | AD (↓) | AG (↑) | FF (↑) | Fid-In (↑) | SPS (↑) | COMP (↓) | MM |
|---|---|---|---|---|---|---|---|---|
| | *ZS classification on ESC50, STFT-Masking, 80.7% accuracy* | | | | | | | |
| ScoreCAM | 29.97 | 12.14 | 8.82 | 0.70 | 0.75 | 0.32 | 12.59 | 0.41 |
| GScoreCAM | 29.64 | 8.56 | 6.62 | **0.79** | 0.84 | 0.36 | 12.52 | 0.39 |
| LMAC-ZS Clotho | 37.40 | 7.43 | **11.26** | 0.78 | 0.86 | 0.50 | 12.29 | 0.11 |
| LMAC-ZS FSD50K | 34.00 | 8.33 | 10.12 | 0.77 | 0.83 | 0.61 | 11.83 | 0.04 |
| LMAC-ZS AudioCaps | 39.00 | 5.93 | 10.43 | 0.78 | 0.88 | **0.68** | **11.67** | 0.07 |
| LMAC-ZS MACs | 15.61 | 22.86 | 5.32 | 0.78 | 0.61 | 0.42 | 12.42 | 0.04 |
| LMAC-ZS Subset (25%) | 41.50 | **3.48** | 7.99 | **0.79** | **0.92** | 0.65 | 11.91 | 0.22 |
| LMAC-ZS Subset (50%) | **43.70** | 3.54 | 7.86 | **0.79** | 0.91 | 0.63 | 11.97 | 0.19 |
| LMAC-ZS Subset (75%) | 40.60 | 4.74 | 7.73 | **0.79** | 0.89 | 0.66 | 11.84 | 0.17 |
| LMAC-ZS All Data | 43.35 | 4.29 | 10.57 | 0.78 | 0.90 | 0.65 | 11.86 | 0.10 |
| | *ZS classification on ESC50, STFT-Masking, ESC50 contamination, 58.6% accuracy* | | | | | | | |
| ScoreCAM | 31.39 | 7.03 | 7.05 | **0.79** | **0.87** | 0.36 | 12.52 | 0.39 |
| GScoreCAM | 28.07 | 13.74 | 8.42 | 0.70 | 0.73 | 0.32 | 12.59 | 0.41 |
| LMAC-ZS Clotho | 35.65 | 12.23 | **13.04** | 0.69 | 0.74 | 0.53 | 12.18 | 0.09 |
| LMAC-ZS AudioCaps | 35.97 | 10.35 | 11.42 | 0.68 | 0.76 | 0.71 | 11.63 | 0.07 |
| LMAC-ZS FSD50K | 26.95 | 16.26 | 9.97 | 0.67 | 0.65 | 0.66 | **11.59** | 0.03 |
| LMAC-ZS MACS | 11.38 | 31.54 | 4.42 | 0.68 | 0.38 | 0.44 | 12.41 | 0.05 |
| LMAC-ZS Subset (25%) | **42.65** | **5.99** | 9.81 | 0.70 | 0.84 | 0.66 | 11.90 | 0.20 |
| LMAC-ZS Subset (50%) | 39.47 | 7.52 | 9.05 | 0.71 | 0.81 | 0.66 | 11.88 | 0.16 |
| LMAC-ZS Subset (75%) | 40.42 | 7.07 | 8.84 | 0.70 | 0.83 | **0.68** | 11.80 | 0.16 |
| LMAC-ZS All Data | 39.47 | 8.28 | 11.81 | 0.69 | 0.80 | 0.67 | 11.79 | 0.09 |

LMAC-ZS is a decoder-based interpreter. That is, we train the decoder based on the pre-trained classifier's representations. The amount and quality of training data can impact both the performance and the training time of the interpreter. We benchmarked our interpreter on different datasets with different sizes, i.e. the datasets that are included within the whole CLAP training set (denoted with All Data in Table 4) - The datasets that make up the whole CLAP training set are, Clotho [37], MACs [40], FSD50k [38] and AudioCaps [39]. We have also experimented with randomly subsampling the whole CLAP training set and denoted it as 'Subset' in 4.

In Table 4, we report the interpreter performance for the aforementioned training datasets with different sizes. We note that the explanation's faithfulness is comparable when training the decoder on the entire training data or a subset; this indicates that it is possible to train LMAC-ZS on a smaller dataset and still obtain faithful explanations.

Table 5: Frechet Audio Distance of the training datasets with respect to ESC-50.

|        | MACs | FSD50k | Clotho | AudioCaps | Subset (25%) |
|--------|------|--------|--------|-----------|--------------|
| ESC-50 | 3.33 | 3.04   | 3.09   | 3.11      | 3.18         |

However, we note that the MACs-only training results obtained the lowest performance on ESC50. We note that this is likely related to the differences in data distributions. To investigate this, we have computed Frechet Audio Distances (computed via CLAP embeddings) between ESC-50 and different subsets in Table 5. We observe that the highest distance is between MACs and ESC50 is the highest. This suggests that if the similarity between the target domain and the training set for the interpreter is relatively high, it is possible to train LMAC-ZS on smaller subsets.

In Table 4, we also present the performance of two additional baselines, ScoreCAM [48] and GScoreCAM [49], and we see that on ESC50, except the case where input audio is contaminated with another audio sample from the ESC50 dataset, LMAC-ZS is able outperform these baselines for the majority of the faithfulness metrics.

