# OpenReview forum: "Listenable Maps for Zero-Shot Audio Classifiers"
_NeurIPS.cc/2024/Conference — NeurIPS 2024 poster_

### Official Review · Reviewer_DqWc · 2024-07-03

**Soundness:** 3
**Presentation:** 3
**Contribution:** 2
**Rating:** 5
**Confidence:** 4

**Summary:**

This paper describes an extension of the LMAC method for explaining decisions made by audio classifiers.  The novelty in the present article is to extend from fixed-vocabulary settings to zero-shot / open text description settings.  To accomplish this, the authors propose a training objective that aims to preserve proximity of (learned) masked audio embeddings to embeddings of corresponding text descriptions.  The method is evaluated on the CLAP model over two standard sound event detection datasets, and compares quite favorably to prior work along several evaluation criteria.

**Strengths:**

Overall, I found this paper to be well written and easy to follow.  The topic of the paper is timely and relevant, as much audio analysis research does seem to be trending toward open vocabulary settings.  The empirical evaluations are appropriate and generally convincing.

The proposed method makes a lot of intuitive sense, and appears to be a natural extension of the fixed vocabulary setting (section 2.2).

**Weaknesses:**

The main weakness I see in this work is a somewhat shallow investigation of the proposed method itself, as opposed to high-level comparisons to prior work.  The core technical contribution is described in equations 5-7, and consists of 3 terms: one to approximately preserve the embedding of audio (relative to text embedding) after masking, one to promote sparsity on the masks, and one to promote diversity of masks when conditioning on different text embeddings.  As in CLAP, contrast is obtained by comparing amongst other samples within the training batch.  This raises a handful of questions about the various components of the training objective which I will detail below; however, the critical issue here is that no ablation study is conducted to measure the importance of the various terms.  This leads the reader at a bit of a loss to understand how these components interact with each other, whether they're all necessary, and so on.

**Questions:**

My questions primarily center around equation 6:

- Notational quirk: $C_{i,j}$ is a scalar value (equation 5), right?  Why are norm bars ($\|$) used in eq 6 when computing the loss on approximating $C_{i,j}$?
- Reasoning through the first term of eq6, a few things jump out at me.  First, expanding out the definition in eq5, the summand should be equivalent to $\left|t_i^\mathsf{T} \left(f_\text{audio}(X_{\text{audio},j}) - f_\text{audio}(M_\theta(t_i, h_j) \odot X_{\text{audio},j}) \right)\right|$.  That is, the gap between the original and masked audio embedding should be small (ie the mask doesn't do much) or approximately orthogonal to $t_i$, and this should hold for all $t_i$ in batch when the outer sum is computed.
    - What is the effect of the batch size / batch diversity on this loss?  Presumably, the more (and more diverse) text embeddings we compute against, the harder it will be to achieve this orthogonality while having nontrivial masks.  Similarly, a too-small batch could make it too easy to achieve this orthogonality in a high-dimensional embedding space while not producing useful masks.  Is there a sweet spot or range sizes where this seems to work (on CLAP)?
- What happens if the third term is left out of the loss?  Similarly, how sensitive is this term to size and diversity of the batch?

A couple of minor questions:

- Line 155 describes the audio reconstruction by ISTFT.  It's not clear how phase is treated here, since $X$ appears to be magnitude spectra.  Presumably phase is retained from the STFT and propagated through, but if some other method is used here (eg griffin-lim or the like) it should be made explicit.
- In general, the spectrogram visualizations (eg figure 4) are quite difficult to read in this paper.  It would be helpful to A) label the axes with appropriate units, B) use a log-scale frequency axis for display (even if the underlying method uses linear spectra, the results will be more visible), and C) use a log amplitude scaling on the magnitudes to enhance visual contrast.

**Limitations:**

The limitations and impacts are well and appropriately described by the authors.

---

> ### Author Rebuttal · Authors · 2024-08-06
>
> We thank you for your comments. Our replies are below:
>
> - Regarding your comment on the lack of ablation study regarding different terms of the loss function:
>
>   We have conducted an ablation study (both quantitative and qualitative) to showcase the relevance of the unimodal diversity loss introduced in Eq. 7.
>
> - Regarding the notational quirk on $C_{ij}$ you make reference to:
>
>   You are correct. We will remove the norm bars in Equation 6.
>
> - Regarding your comment on the first term in equation 6, and whether a trivial, all-ones mask would minimize this loss function:
>
>   It definitely would. Because of this, the second term in Equation 6 is very important, and as we indicate in line 142 of the submission, this term is needed to avoid trivial solutions (all-ones mask) :  `The second term in Equation 6 promotes sparsity in the generated mask to avoid trivial solutions`.
>
> - Regarding your comment,
>   > What happens if the third term is left out of the loss?
>
>   The third term in the loss function is responsible for making the decoder more sensitive to text prompts. To validate its impact on the generated explanations, we have carried out a qualitative and quantitative ablation study for this. Please see the qualitative results in the .pdf document attached for the rebuttal. The quantitative comparison indicates that by introducing the third loss term we marginally decrease the explanation’s faithfulness, while increasing the sensitivity to text prompts significantly (Fig 3 in the attached pdf).
>
> - > Similarly, how sensitive is this term (third term) to size and diversity of the batch?
>
>   First of all, note that the proposed loss function is not contrastive. It is rather a loss term that matches two matrices. So, the behavior is expected to be more similar to regular loss functions. To quantify the effect of batch size on the end performance of the model, we have also trained a model with BS=4 on the AudioCaps dataset and reported the results in Table 5. We observe that with BS=4 in fact the faithfulness numbers are slightly worse compared to BS=2 (potentially due to suboptimal hyperparameters).
>
>
> - Regarding your comment,
> > Line 155 describes the audio reconstruction by ISTFT. It's not clear how phase is treated here, since appears to be magnitude spectra. Presumably phase is retained from the STFT and propagated through, but if some other method is used here (eg griffin-lim or the like) it should be made explicit.
>
>   The phase of the original input audio is used to reconstruct the listenable audio for the explanation (same way it is done in the original L-MAC paper, equation 3.) We will clarify this in the final version of the paper.
>
> - Regarding your comments,
> > In general, the spectrogram visualizations (eg figure 4) are quite difficult to read in this paper. It would be helpful to A) label the axes with appropriate units, B) use a log-scale frequency axis for display (even if the underlying method uses linear spectra, the results will be more visible), and C) use a log amplitude scaling on the magnitudes to enhance visual contrast.
>
>   Thank you for your comment. We will use a log-amplitude scaling on the magnitudes / log-scale frequency axis for display, and label the axes.

---

> > ### Comment · Reviewer_DqWc · 2024-08-09
> >
> > Thanks for your responses - these address all of my questions above.

---

> ### Author Response · Authors · 2024-08-09
>
> Thanks a lot for your positive feedback on our rebuttal! Please consider raising your score as it might help with the final decision. Thank you very much consideration.

---

### Official Review · Reviewer_vLBn · 2024-07-03

**Soundness:** 3
**Presentation:** 3
**Contribution:** 3
**Rating:** 6
**Confidence:** 4

**Summary:**

The paper introduces a post-hoc interpretation method for zero-shot audio classifiers, named LMAC-ZS (Listenable Maps for Audio Classifiers in the Zero-Shot context). It addresses the challenge of interpreting predictions from zero-shot audio classifiers that define audio classes based on textual prompts, where labels are not predefined but generated dynamically. LMAC-ZS outputs saliency maps that highlight important regions within input audio, correlating these with corresponding text prompts. The method involves a novel loss function that maintains the original audio-text similarity, enhancing interpretability. Experiments were conducted using the CLAP model on standard datasets like ESC50 and UrbanSound8K.

**Strengths:**

1. Motivation is Clearly articulated. It stresses the importance of interpretability in AI, particularly for models used in critical decision-making areas, focusing on the less-explored zero-shot audio classification.
2. The problem of providing interpretable explanations for zero-shot audio classifiers is clearly formulated and identified as a novel challenge in the field.
3. The paper compares LMAC-ZS with several baseline methods like GradCAM++, SmoothGrad, and Integrated Gradients, providing a thorough comparative analysis and justifying the selection based on their relevance and common use in related tasks.

**Weaknesses:**

1. It is observed that in some scenarios LMAC-ZS (CT) performs better than LMAC-ZS (FULL) in metrics such as AI, AD, and AG. This is counterintuitive, as one would expect the model trained on the full CLAP dataset to perform better. An explanation for this discrepancy is needed.
2. The paper mentions exploring the training of LMAC-ZS only on the Clotho dataset to simulate a limited computational budget scenario. It raises the question of whether training on other individual datasets or comparing the performance across different single datasets versus the full dataset would yield different insights. This aspect needs further exploration and clarification.
3. In Table 2, the use of ESC50 contamination for ZS classification on the ESC50 dataset seems confusing. The rationale behind using the same dataset for contamination needs to be explained in detail. Similarly, in Table 3, the reason for using US8K contamination for Mel-Masking and ESC50 contamination for STFT-Masking in ZS classification on US8K requires clarification.

**Questions:**

1. In Figure 7, the visual explanations for 'Toilet flushing' and 'Water drops' look very similar despite different similarity scores (Sim=0.69 and Sim=0.14). How does the model ensure distinguishable and meaningful explanations, and can more distinct examples be provided to demonstrate the method's effectiveness?
2. I have a question about a set of textual prompts. What happens if there are sounds that are not in the text?

**Limitations:**

The author has addressed the limitations.

---

> ### Author Rebuttal · Authors · 2024-08-06
>
> We thank you for your constructive and positive comments. Our replies are below:
>
> - Regarding your comment as to the discrepancy between the L-MAC-ZS Full and L-MAC-ZS (CT) in some cases:
>
>   This is potentially due to optimization, and choice of hyperparameters (note also that this is mainly observed for mel-masking and not STFT masking). From preliminary results on training L-MAC-ZS (Full) for few more epochs for mel-masking, and we see that the results seem to be on par.
>
> - Regarding your comment on training L-MAC-ZS with other subsets (subsets other than Clotho):
>
>   In Tables 3 and 4 (the general reply above) we report the results obtained by training the decoder on different subsets. We experimented both with randomly sampled subsets of the whole CLAP dataset and with the individual datasets comprising the CLAP training data (namely AudioCaps, FSD50K, MACs, and Clotho).  As mentioned above, we note that the explanation’s faithfulness is comparable when training the decoder on the full training data or a subset. We note that the training with only MACs dataset results in the lowest performance on ESC50. We note that this is likely related to the differences in data distributions. In fact, the MACs / ESC-50 Frechet Audio Distance (computed using CLAP embeddings) is the highest among all the subsets (Table 6 in the general rebuttal above).  We conclude that training on a subset of the training data, the decoder can generate faithful explanations for the ZS classifier.
>
> - Regarding your comment:
>  > In Table 2, the use of ESC50 contamination for ZS classification on the ESC50 dataset seems confusing.
>
>   This setting refers to the case where we create sound mixtures using two recordings with different classes from ESC50. Same is true for the US8k dataset as well. The goal is to investigate whether L-MAC-ZS is able to point out the important parts of the recording in a more complicated audio recording.
>
> - Regarding your comment:
> > In Figure 7, the visual explanations for 'Toilet flushing' and 'Water drops' look very similar despite different similarity scores (Sim=0.69 and Sim=0.14). How does the model ensure distinguishable and meaningful explanations, and can more distinct examples be provided to demonstrate the method's effectiveness?
>
>   We have provided additional qualitative examples in the .pdf document provided for the general rebuttal.
>
> - Regarding your question
>   > I have a question about a set of textual prompts. What happens if there are sounds that are not in the text?
>
>   In this case, as demonstrated in the example in Figure 4 and more generally Figure 3, L-MAC-ZS is able to return an empty mask when the similarity between the text prompt and the input audio is low. We have provided additional evidence to this in the pdf attached to the general rebuttal.

---

> > ### Comment · Reviewer_vLBn · 2024-08-09
> >
> > Thank you for addressing the questions! I am satisfied with the responses and have no further questions at this time.

---

### Official Review · Reviewer_Tsn7 · 2024-07-08

**Soundness:** 3
**Presentation:** 1
**Contribution:** 2
**Rating:** 6
**Confidence:** 4

**Summary:**

In this paper, the authors focus on interpreting the decisions of zero-shot audio classifiers, particularly ones based on the Contrastive Language-Audio Pretraining (CLAP) model. To achieve this, the authors propose to learn a decoder that predicts a "listenable" audio saliency map (a mask on the input spectrogram) from an audio-text pair. The decoder is trained with a novel loss function that remains faithful to the characteristics of CLAP features. The authors demonstrate the effectiveness of their approach with extensive experiments across various datasets.

**Strengths:**

### Strengths

1. Although there is a significant body of research focusing on interpretability of audio classifiers, interpretability of *zero-shot* audio classifiers (and zero-shot classifiers in general) is an under-explored area, which certainly merits more investigation.
2. From a technical perspective, the proposed approach is sound and has no glaring weaknesses.
3. The qualitative and quantitative results show that the method indeed works as claimed.
    - Particularly the anonymized qualitative samples provided are highly appreciated.

**Weaknesses:**

### Weaknesses

1. Even after reading two out of the three subsections in Section 2: Methodology, a reader learns nothing new from the paper.

    To be more specific:
    - Section 2.1 provides a high-level overview of "Contrastive Language Audio Pretraining" (CLAP) [10], while Section 2.2 gives a brief summary of the approach of the paper "Listenable Maps for Audio Classification" (L-MAC) [8]. Both these sections discuss the previous papers as is, without any new perspective.
    - It is solely Section 2.3 that pertains to the method proposed by the paper. Even in Section 2.3, the sub-subsection @line-152, **"Producing Listenable Interpretations"** is not something new achieved by the proposed method, but rather a feature of the previous L-MAC paper [8].
        - Notably, [8] also has an explicit sub-section with the *exact same title*; **Section 2.2: Producing Listenable Interpretations**. The authors make no effort in lines 152-156 to clarify that the "listenable" feature comes from [8]; thus this is not only redundant, but also very close to plagiarism.

    - **Suggestion:** I strongly suggest that Sections 2.1, 2.2, and the sub-subsection (@lines 152-156) be discussed separately as a "Background" or "Preliminaries" section, instead of Methodology. Specifically, since the proposed approach builds significantly upon L-MAC [8], it is necessary to disentangle your contributions from that of the authors of [8].

2. The technical novelty is limited by the L-MAC paper.
    - From my understanding, the only novelty in the method is the loss function in Equation 6. And even that is mostly changing the cross entropy-based objectives in L-MAC (Equation (2) in [8]) to contrastive losses characteristic of CLAP. Overall, the same min-max objective is retained, and the same regularization term is added for mask sparsity. The same general framework from L-MAC is followed: a decoder is trained on the same dataset as the encoder to predict a mask on the input audio spectrogram.

3. (minor weakness) Comparison against baselines.

    - The authors are comparing against the baselines of GradCAM, GradCAM++, SmoothGrad, Integrated Gradients. While all these methods were also used for comparison with the original L-MAC in [8], that was in a regular classification setting; these methods are, by nature, poorly suited to zero-shot settings.

    - For instance, for both "cat" and "glass-breaking", the CLAP model needs to "look" at the important regions of the audio to give an encoded audio feature---which can be observed in Figure 4 (for GradGAM) in the paper.

    - **Suggestion**: A better baseline would perhaps be from the paper "gScoreCAM: What objects is CLIP looking at?" [a]. The paper empirically establishes that for CLIP, in zero-shot settings, ScoreCAM performs better than the other forms of CAM.

    - *Note:* I understand that in general, there is a lack of baselines to compare to (for instance, listen-to-interpret and L-MAC are not applicable). I do not expect the authors to conduct any additional experiments.

---

In general, the paper has a significant amount of similarity/redundancy with the L-MAC paper [8], from the naming of sections and content (noted in Weakness 1), figures (Fig 1 in [8] and fig 2 in current paper), and even the exact set of metrics posed in the identical order in Section 3.1, which makes it difficult to discern the novelty and contributions.

---
[a] Chen, Peijie, et al. "gscorecam: What objects is clip looking at?." Proceedings of the Asian Conference on Computer Vision. 2022.

**Questions:**

### Questions
1. **line 147** *"The intuition is that the similarity between two text prompts should be reflected in the similarity of the audio embeddings from the corresponding masked spectrograms"*.

    Equation (7) is the only part of the loss function that is distinctly novel, as it adds a third term to the original loss in [8]. Have you tried training the overall decoder without the third term? It would be helpful to the paper if you can show that adding the third term yields a noticeable improvement over just using the first two terms; otherwise, the hypothesis remains unsupported by evidence.

2. It is common to perform zero-shot classification with foundation models trained on a large dataset. Suppose we have one such hypothetical model, trained on a dataset with millions of samples.
    - Is it necessary to train the decoder on the same large dataset? If not, can an estimation be made as to what percentage of the training data the decoder needs to see to achieve reliable performance? (For instance, it may be the case that after seeing 20% of the data, the model achieves 80% of its full performance).
    - If it is necessary to train the decoder on the pre-training dataset, using LMAC-ZS becomes expensive in many cases. Can LMAC-ZS be jointly learned during the pre-training process of the base model (e.g. CLAP)?
        - Note that this may not be so straightforward, as the first term in the loss objective will try to match the decoder predictions to faulty entries of $C$ at the start of the training.

### General Suggestions
- **line-114**; **line-122**: The authors note that they omitted a part of Equation (4) for brevity. I strongly suggest against doing this; *clarity* is more important than brevity when posing a loss function or optimization objective.

    Equation (4) is supposed to be a min-max objective; the goal is to maximize the classification confidence of the masked-in (salient) part of the audio, while minimizing the confidence of the masked-out part. Thus the whole equation should be written together, and parts of it should not be omitted. If brevity is desired, it may be achieved by abbreviating $\text{CrossEntropy}$ to $CE$ or $\mathcal{L}_{CE}$.

- **line 120:** The abbreviation L-MAC is used, but it has not been defined previously. The authors should define the abbreviation on line 70.
- **line 62-63:** The citation style is inconsistent. For example, in lines 59-60, it has been written "Key approaches in this category include [19, 20, 21, 22],". The same citation style can be followed in line 64: "Notable attempts in this vein include [23, 24, 25]."

---
[b] Shimada, Kazuki, et al. "Zero-and Few-Shot Sound Event Localization and Detection." ICASSP 2024-2024 IEEE International Conference on Acoustics, Speech and Signal Processing (ICASSP). IEEE, 2024.

**Limitations:**

### Limitations
- One limitation that I feel is not addressed is that the approach needs to be specifically trained on CLAP's data; it is not plug-and-play like Grad-CAM or similar approaches.
    - In **line 216** It is mentioned that the decoder uses CNN14 layers, presumably because the audio part of CLAP is based on CNN14.
        - Now if we have another zero-shot audio classifier, LAION-CLAP, that (suppose) has the same dataset but a different transformer-like architecture, then the decoder may not be transferable. Another decoder architecture needs to be designed to suit the alternative zero-shot classifier. So for different zero-shot foundation models, it may become necessary to have different architectures, which can be a hurdle.
        - As noted earlier, for same architecture but different datasets, it is still needed to train the decoder again. Unless of course, the decoder from one pre-training dataset transfers to another dataset, as a general purpose audio method.

Apart from these scalability concerns, I believe the remaining limitations are adequately addressed.

---

Update: The rebuttal addresses most of my concerns, so I will raise my score to 6.

---

> ### Author Rebuttal · Authors · 2024-08-06
>
> We thank you for your comments. Our replies are below:
>
> - Regarding your comments on the organization of the paper, and the fact that the paper contains sections pertaining to L-MAC and CLAP:
>
>     Our aim was to make this submission as self-contained as possible, and to provide the reader with the necessary preliminaries. We agree with your suggestion, and we will put Sections 2.1 and 2.2 under a new section called ‘Preliminaries’. We have also noted your comment on the subsection called ‘Producing Listenable Interpretations’, and we will clarify the overlaps with the original L-MAC paper.
>
> - Regarding your comment on limited novelty on top of L-MAC:
>
>   This submission is an extension of the L-MAC framework to zero-shot classifiers, and hence it is called L-MAC-ZS. There are obvious similarities, like L-MAC, L-MAC-ZS is also a masking method that uses a decoder. However, beyond using a decoder network, using masking, and the fact that the masking is done in the STFT domain (to be able to produce listenable explanations) this paper addresses another, and more challenging problem compared to the original L-MAC paper.
>
>   We would like to also state that, to the best of our knowledge, this is the first method that explores using a decoder based method to explain an audio foundation model like CLAP. Also, note that the loss function is significantly different from L-MAC. In L-MAC, the loss function tries to maximally retain logit similarity for mask-in, minimize the logit similarity for mask-out.  In L-MAC-ZS however the interpreter is not trained with data with a fixed set of labels, and the loss function aims to match the original similarities between the text prompts and the audios in the batch, with the similarities obtained after masking. We also introduced a unimodal similarity term in Eq. 7, which we show is important for our results in this rebuttal (Tables 1 and 2 in the general rebuttal) .
>
> - Comparisons with baselines:
>
>    We thank you for your suggestion to compare with gScoreCAM. We have added a comparison with gScoreCAM and ScoreCAM in Tables 1 and 2 of the general rebuttal. We observe that both gScoreCAM and ScoreCAM faithfulness scores are lower than L-MAC-ZS.
>
>
> - Regarding your question whether using the third term (given in equation 7) in the loss function is useful:
>
>    We have conducted a qualitative (the rebuttal .pdf file) and quantitative ablation study (Tables 1 and 2 in the general rebuttal). We observe that by introducing the unimodal alignment loss term explanations become sensitive to the text prompt (Figure 3 in the pdf) while maintaining comparable faithfulness metrics (Tables 1 and 2). This is confirmed qualitatively as we note that explanations generated without the unimodal diversity term remain unchanged regardless of the provided text prompt.
>
>
> - Regarding your question on if we need to train the decoder on the same dataset as the foundation model to be interpreted (e.g. CLAP):
>
>    Please note that we had partially investigated this in the submission. In Tables 1, 2, 3 of the original manuscript we have provided results with a decoder that had only been trained with Clotho (approximately ¼ of the whole training dataset). We see that results do not significantly drop. To further investigate this, we conducted an ablation study on the amount of data needed to train the decoder (Tables 3 and 4 in the general rebuttal). We experimented with randomly sampled subsets of the CLAP dataset and with the individual datasets comprising the CLAP training data (namely AudioCaps, FSD50K, MACs, and Clotho). As observed in the submission, we note that the explanation’s faithfulness is comparable when training the decoder on the full training data or a subset. The MACs-only training results in the lowest performance on ESC50. We note that this is likely related to the differences in data distributions. In fact, the MACs / ESC-50 Frechet Audio Distance (computed using CLAP embeddings) is the highest among all the subsets (Table 6). We conclude that training on a subset of the training data, the decoder can generate faithful explanations for the ZS classifier.
>
>
> - > If it is necessary to train the decoder on the pre-training dataset, using LMAC-ZS becomes expensive in many cases. Can LMAC-ZS be jointly learned during the pre-training process of the base model (e.g. CLAP)?
>
>    In this paper we have not experimented with jointly training CLAP and L-MAC-ZS. However, this is an interesting future direction. Also, please note that in response to your earlier comment, it is also possible to train LMAC-ZS on a subset of the pre-training dataset.
>
> - Regarding your comment,
> > line-114; line-122: The authors note that they omitted a part of Equation (4) for brevity. I strongly suggest against doing this; clarity is more important than brevity when posing a loss function or optimization objective.
>
>    We will include the full loss function in Equation 4.
>
> - Regarding your comments on citation style consistency, and undefined abbreviation in line 120:
>
>   We will fix these issues in the final version of the paper.
>
> - Regarding your comment:
> > One limitation that I feel is not addressed is that the approach needs to be specifically trained on CLAP's data; it is not plug-and-play like Grad-CAM or similar approaches.
> and your related comments regarding generalizability to different foundation model architectures, and different datasets:
>
>    Generalizability to different audio encoder architectures: There’s no theoretical reason as to why the convolutional decoder would not provide faithful and understandable explanations for different foundation model architectures. However, it also remains feasible to design new decoder architectures based for a different foundation model. Regarding different training datasets: As we mentioned earlier above, we have experimented with training the decoder with subsets of the training data, and we see that the results remain comparable.

---

> > ### Comment · Reviewer_Tsn7 · 2024-08-07
> >
> > Thank you, my primary concerns have been addressed. I will raise my score.

---

> ### Author Response · Authors · 2024-08-07
>
> Thank you for your detailed review, and your prompt response!

---

### Author Rebuttal · Authors · 2024-08-06

We thank all the reviewers for their comments. We provide a reply to each reviewer in the corresponding rebuttal. Here, we report the additional quantitative results obtained during the rebuttal period to address the reviewer's concerns. The reviewer replies refer to the table numbers listed below.

**Table 1.**  In-Domain Results on ESC-50, **LMAC-ZS (CT)** refers to LMAC-ZS trained on Clotho (with the diversity loss term), **LMAC-ZS (CT) NoDiv** refers to LMAC-ZS trained on Clotho without the diversity term (Eq. 7 in the paper). We only show the baselines that perform close to LMAC-ZS.

| **Metric**                  | AI (↑) | AD (↓) | AG (↑) | FF (↑) | Fid-In (↑) | SPS (↑) | COMP (↓) | MM   |
|-----------------------------|--------|--------|--------|--------|------------|---------|----------|------|
| GradCam                     | 20.30  | 23.75  | 7.77   | 0.78   | 0.58       | 0.72    | 11.54    | 0.14 |
| GradCam++                   | 32.50  | 8.97   | 7.95   | 0.79   | 0.84       | 0.41    | 12.41    | 0.35 |
| ScoreCAM                    | 29.97  | 12.14  | 8.82   | 0.70   | 0.75       | 0.32    | 12.59    | 0.41 |
| GScoreCAM                   | 29.64  | 8.56   | 6.62   | 0.79   | 0.84       | 0.36    | 12.52    | 0.39 |
| **LMAC-ZS (CT)**            | 37.40  | 7.43   | 11.26 | 0.78   | 0.86       | 0.50    | 12.29 | 0.11 |
| **LMAC-ZS (CT) NoDiv**      | 37.54  | 6.38 | 11.70| 0.77   | 0.88   | 0.72    | 11.59 | 0.02 |

**Table 2.** Out-of-Domain Results on ESC50 Mixtures

| **Metric**                  | AI (↑) | AD (↓) | AG (↑) | FF (↑) | Fid-In (↑) | SPS (↑) | COMP (↓) | MM   |
|-----------------------------|--------|--------|--------|--------|------------|---------|----------|------|
| GradCam                     | 23.77  | 25.25  | 12.24  | 0.69   | 0.49       | 0.69    | 11.73    | 0.17 |
| GradCam++                   | 29.52  | 14.84  | 10.17  | 0.70 | 0.70       | 0.39    | 12.48    | 0.35 |
| ScoreCAM                    | 31.39  | 7.03 | 7.05   | 0.79   | 0.87   | 0.36    | 12.52    | 0.39 |
| GScoreCAM                   | 28.07  | 13.74  | 8.42   | 0.70   | 0.73       | 0.32    | 12.59    | 0.41 |
| **LMAC-ZS (CT)**            | 35.65  | 12.23  | 13.04  | 0.69   | 0.74       | 0.53    | 12.18    | 0.09 |
| **LMAC-ZS (CT) NoDiv**      | 37.35 | 9.55   | 13.53 | 0.67   | 0.79       | 0.75 | 11.46 | 0.02 |


**Table 3.** Ablation of the amount of training data (the model is trained only on the indicated dataset) - In-Domain Setting (ESC50)

| **Metric**                  | AI (↑) | AD (↓) | AG (↑) | FF (↑) | Fid-In (↑) | SPS (↑) | COMP (↓) | MM   |
|-----------------------------|--------|--------|--------|--------|------------|---------|----------|------|
| LMAC-ZS Clotho          | 37.40  | 7.43   | 11.26  | 0.78   | 0.86       | 0.50    | 12.29    | 0.11 |
| LMAC-ZS FSD50K          | 34.00  | 8.33   | 10.12  | 0.77   | 0.83       | 0.61    | 11.83    | 0.04 |
| LMAC-ZS AudioCaps       | 39.00  | 5.93   | 10.43  | 0.78   | 0.88       | 0.68    | 11.67    | 0.07 |
| LMAC-ZS MACs            | 15.61  | 22.86  | 5.32   | 0.78   | 0.61       | 0.42    | 12.42    | 0.04 |
| LMAC-ZS Subset (25%)    | 41.50  | 3.48   | 7.99   | 0.79   | 0.92       | 0.65    | 11.91    | 0.22 |
| LMAC-ZS Subset (50%)    | 43.70  | 3.54   | 7.86   | 0.79   | 0.91       | 0.63    | 11.97    | 0.19 |
| LMAC-ZS Subset (75%)    | 40.60  | 4.74   | 7.73   | 0.79   | 0.89       | 0.66    | 11.84    | 0.17 |
| LMAC-ZS CLAP Data       | 43.35  | 4.29   | 10.57  | 0.78   | 0.90       | 0.65    | 11.86    | 0.10 |

**Table 4.** Ablation of the amount of training data (the model is trained only on the indicated dataset)  - Out-of-Domain Setting (ESC50 Mixtures)

| **Metric**                  | AI (↑) | AD (↓) | AG (↑) | FF (↑) | Fid-In (↑) | SPS (↑) | COMP (↓) | MM   |
|-----------------------------|--------|--------|--------|--------|------------|---------|----------|------|
| LMAC-ZS Clotho          | 35.65  | 12.23  | 13.04  | 0.69   | 0.74       | 0.53    | 12.18    | 0.09 |
| LMAC-ZS AudioCaps       | 35.97  | 10.35  | 11.42  | 0.68   | 0.76       | 0.71    | 11.63    | 0.07 |
| LMAC-ZS FSD50K          | 26.95  | 16.26  | 9.97   | 0.67   | 0.65       | 0.66    | 11.59    | 0.03 |
| LMAC-ZS MACS            | 11.38  | 31.54  | 4.42   | 0.68   | 0.38       | 0.44    | 12.41    | 0.05 |
| LMAC-ZS Subset (25%)    | 42.65  | 5.99   | 9.81   | 0.70   | 0.84       | 0.66    | 11.90    | 0.20 |
| LMAC-ZS Subset (50%)    | 39.47  | 7.52   | 9.05   | 0.71   | 0.81       | 0.66    | 11.88    | 0.16 |
| LMAC-ZS Subset (75%)    | 40.42  | 7.07   | 8.84   | 0.70   | 0.83       | 0.68    | 11.80    | 0.16 |
| LMAC-ZS CLAP Data       | 39.47  | 8.28   | 11.81  | 0.69   | 0.80       | 0.67    | 11.79    | 0.09 |

**Table 5.** Ablation on the batch size

| **Metric**                  | AI (↑) | AD (↓) | AG (↑) | FF (↑) | Fid-In (↑) | SPS (↑) | COMP (↓) | MM   |
|-----------------------------|--------|--------|--------|--------|------------|---------|----------|------|
| AudioCaps BS=4 OOD          | 27.15  | 18.00  | 10.10  | 0.66   | 0.63       | 0.70    | 11.75    | 0.03 |
| AudioCaps BS=2 OOD          | 35.97  | 10.35  | 11.42  | 0.68   | 0.76       | 0.71    | 11.63    | 0.07 |
| AudioCaps BS=4 ID           | 30.00  | 12.03  | 8.92   | 0.76   | 0.78       | 0.64    | 11.75    | 0.06 |
| AudioCaps BS=2 ID           | 39.00  | 5.93   | 10.43  | 0.78   | 0.88       | 0.68    | 11.67    | 0.07 |

**Table 6.** Frechet Audio Distance between ESC-50 and the datasets, computed using CLAP representations
| Comparison                 | FAD |
|----------------------------|-------|
| MACs               | 3.33  |
| FSD50K                  | 3.04  |
| Clotho               | 3.09  |
| AudioCaps            | 3.11  |
| CLAP Subset (25%)    | 3.18  |

---

### Decision · Program_Chairs · 2024-09-25

**Decision:**

Accept (poster)

**Comment:**

All reviewers agreed that the paper has merit, with the ever increasing need of interpretability in audio tasks. There were a few concerns in the experiments, e.g., raised by reviewer vLBn and DqWc, but they were resolved during the rebuttal. The authors also provide an extensive comparison in the rebuttal to showcase the performance of their approach.

I'm sharing some of the concerns from reviewer Tsn7. The paper reads too similar to Paissan et al. (2024). I'm surprised how little contrast has been made between this paper and Paissan et al. (2024). Given how similar the approaches are, the paper should make it absolutely clear that the idea (and even the experimental design) is taken from Paissan et al. (2024) and is extended it to the zero-shot setting. This will help make the novelty clear. Given the rebuttal, I'm confident that the authors will make the best effort to improve the writing and giving more credits to Paissan et al. (2024).